



# Ion distribution functions in magnetotail reconnection: Global hybrid-Vlasov simulation results

Andrei Runov[1], Maxime Grandin[2], Minna Palmroth[2,3], Markus Battarbee[2], Urs Ganse[2], Heli Hietala[4,5], Sanni Hoilijoki[6], Emilia Kilpua[2], Yann Pfau-Kempf[2], Sergio Toledo-Redondo[7,8], Lucile Turc[2], and Drew Turner[9]

[1]Institute of Geophysics and Planetary Physics, University of California at Los Angeles, Los Angeles, USA
[2]University of Helsinki, Department of Physics, Helsinki, Finland
[3]Space and Earth Observation Centre, Finnish Meteorological Institute, Helsinki, Finland
[4]The Blackett Laboratory, Imperial College, London, UK
[5]Department of Physics and Astronomy, University of Turku, Turku, Finland
[6]Laboratory for Atmospheric and Space Physics, University of Colorado at Boulder, Boulder, USA
[7]Department of Electromagnetism and Electronics, University of Murcia, Murcia, Spain
[8]Institut de Recherche en Astrophysique et Planétologie, Toulouse, France
[9]The Johns Hopkins University Applied Physics Laboratory, Laurel, MD, USA

**Correspondence:** Andrei Runov (arunov@igpp.ucla.edu)

**Abstract.** We present results of noon–midnight meridional plane global hybrid-Vlasov simulations of the magnetotail ion dynamics under steady southward interplanetary magnetic field using the Vlasiator model. The simulation results show magnetotail reconnection and formation of earthward and tailward fast plasma outflows. The hybrid-Vlasov approach allows us to study ion velocity distribution functions (VDFs) that are self-consistently formed during the magnetotail evolution. We examine the VDFs collected by virtual detectors placed along the equatorial magnetotail within earthward and tailward outflows and around the quasi-steady X-line formed in the magnetotail at $X \approx -14\,R_{\mathrm{E}}$. This allows us to follow the evolution of VDFs during earthward and tailward motion of reconnected flux tubes as well as study signatures of unmagnetized ion motion in the weak magnetic field near the X-line. The VDFs indicate actions of Fermi-type and betatron acceleration mechanisms, ion acceleration by the reconnection electric field, and Speiser-type motion of ions near the X-line. The simulated VDFs are compared and show good agreement with VDFs observed in the magnetotail by the Time History of Events and Macroscale Interactions during Substorms (THEMIS) and Acceleration, Reconnection, Turbulence and Electrodynamics of Moon's Interaction with the Sun (ARTEMIS) spacecraft. We find that the VDFs become more gyrotropic but retain transverse anisotropy and counter-streaming ion beams when being convected earthward. The presented global hybrid-Vlasov simulation results are valuable for understanding physical processes of ion acceleration during magnetotail reconnection, interpretation of in-situ observations, and for future mission development by setting requirements on pitch-angle and energy resolution of upcoming instruments.



# 1 Introduction

According to the near-Earth neutral line model (e.g., Baker et al., 1996), magnetotail reconnection occurs at geocentric distances of 20–30 Earth radii ($R_{\mathrm{E}}$) (Nagai et al., 1998, 2005), and may operate in the fast, impulsive regime as well as in the quasi-steady regime. The quasi-steady reconnection regime may involve formation of magnetic islands or flux-ropes (O-points), separated by multiple X-lines (e.g., Slavin et al., 1995). Generally, tail reconnection produces earthward and tailward outflows carrying northward and southward magnetic fields, respectively, which have been reported in a number of observational studies (e.g., Petrukovich et al., 1998; Angelopoulos et al., 2008a, 2013; Oka et al., 2011; Runov et al., 2012). Observations suggest that fast reconnection outflows are accompanied by strong enhancements in the north-south magnetic field component ($B_z$). The $B_z$ enhancements associated with the earthward outflows typically exhibit a sharp increase at the leading edge referred to as the dipolarization front (Nakamura et al., 2002; Runov et al., 2009), followed by a gradual $B_z$ decrease. The entire magnetic structure associated with the outflow is referred to as the dipolarization flux bundle (DFB) (Liu et al., 2013). On the tailward side of fast reconnection, the negative $B_z$ variation includes a similar structure, with a front on the leading edge (Li et al., 2014; Angelopoulos et al., 2013).

Reconnection affects the particle characteristics in many ways. Generally, reconnection is a process converting electromagnetic energy stored in magnetic fields into the kinetic and thermal energies of the plasma. The modifications in particle kinetics translate into characteristic changes in particle distribution functions, which show particle phase-space density as a function of velocity or energy, that are collected in-situ by particle detectors on board spacecraft (e.g., Ashour-Abdalla et al., 1996). Interpretation of the observed distribution functions is, however, challenging because particles of different origin and history hit the detector during the sampling time. Kinetic modeling is required to understand the underlying physics of particle energization during reconnection. Full kinetic particle-in-cell (PIC) simulations are able to describe kinetics of both ions and electrons with a limited $m_i/m_e$ ratio and within a limited spatial domain. Despite these limitations, velocity distribution functions (VDFs) obtained in PIC models of magnetotail reconnection have revealed patterns similar to observed ones. For example, Hoshino et al. (1998) studied ion VDFs (in the context of this paper, we will further focus on the ion kinetics) observed by the Geotail satellite and compared them with results of two-dimensional PIC simulations. They showed that the ion VDFs near and in the reconnection region are non-Maxwellian and include counter-streaming beams, nongyrotropic and thermalized populations.

Hybrid models, which describe ions as particles and electrons as a massless charge-neutralizing fluid, have been shown capable of resolving ion kinetics in a spatial domain large enough to be regarded as a realistic magnetotail configuration. Krauss-Varban and Omidi (1995) conducted 2D hybrid modeling of reconnection of the current sheet with anti-parallel magnetic field $B_x(z)$ and $B_y = B_z = 0$ (the conventional magnetotail coordinates with $\mathbf{e}_x$ directed toward towards Sun, $\mathbf{e}_y$ duskward, and $\mathbf{e}_z$ along the Earth's dipole was used), which may represent the distant magnetotail, in a simulation box $120\,R_{\mathrm{E}} \times 20\,R_{\mathrm{E}}$. The simulations showed that ion VDFs changed from a non-gyrotropic distribution at large $B_x$ to a gyrotropic one as $B_x \to 0$. Lin and Swift (1996) extended this modeling adding the guide field ($B_y \neq 0$) into the initial setup. They also reported significant changes in the ion distribution function with distance from $B_x = 0$ with strong parallel anisotropy at large $B_x$. The increase in parallel temperature at large $B_x$ was interpreted as a result of ion acceleration by slow shocks. Ion kinetics in a 2D hybrid





reconnection model was studied by Scholer and Lottermoser (1998) and Lottermoser et al. (1998) who found that ion VDFs are structured and non-Maxwellian everywhere in the reconnecting current sheet. Specifically, they identified counterstreaming beam and partial ring distributions.

A number of simulations showed that near the neutral line ions are unmagnetized and perform a complex motion known as Speiser (after Speiser, 1965) motion (e.g., Nakamura et al., 1998; Arzner and Scholer, 2001). Farther toward the reconnection outflows partial-shell ion VDFs were detected. Within the reconnection outflows, ion VDFs were counterstreaming.

Particle energization due to reconnection is not limited to the reconnection site. The reconnection outflows with embedded magnetic structures (DFBs or flux ropes) carry magnetic flux and are represented as localized enhancements in the cross-tail electric field $E_y$. Ambient plasma sheet ions interact with the DFB-associated electric fields and experience magnetic mirror-type acceleration gaining energy from the moving dipolarization front. This effect was identified in observations and modeled using a simple non-self-consistent particle-tracing model with prescribed electric and magnetic fields (Zhou et al., 2010, 2012). While these simple test-particle modeling results do reproduce the main characteristics within observations, modeling of this effect with a self-consistent ion kinetic model is outstanding.

Energetic ions populating the near-Earth plasma sheet may also be picked up by the electric field at and behind the front and be transported toward the inner magnetosphere (e.g., Birn et al., 2015; Runov et al., 2015). Observations show that in these situations the VDFs develop an energy-dependent anisotropy. Specifically, the subthermal ion population is field-aligned, often forming two counterstreaming beams (Eastwood et al., 2015), whereas phase-space density of ions with velocities $v$ as compared to the thermal velocity $V_{th}$ appearing in the range $V_{th} < v < 3V_{th}$ is larger in the field-transversal direction (Runov et al., 2017). The low-energy field-aligned beams and some indications of the field-transversal anisotropy at higher energies were found in PIC simulations (Pritchett and Runov, 2017). The ion VDFs with energy-dependent anisotropy were also modeled using test particle tracing utilizing MHD-simulated electromagnetic fields (Birn and Runov, 2016), showing that the counterstreaming low-energy ion beams are formed by freshly reconnected particles following the reconnecting field line. Because of rapid earthward contraction of the reconnected field line, those particles are subject to Fermi-type acceleration. This field-transversal anisotropy of energetic particles is thus a result of two effects: the betatron energization of local particles due to increase in the magnetic field strength (dipolarization) and the pick-up of energetic particles from the ambient plasma sheet that experience additional energization due to convection toward stronger magnetic field.

Due to non-self-consistency, these results obtained with the test-particle approach are limited. A proof with models that resolve ion kinetics self-consistently in a setup including the reconnection site as well as reconnection outflows at macroscopic scales is required. Further, the particle VDF modeling results are achieved by local simulations describing only the tail reconnection site. Processes of particle energization and transport are different in the earthward outflow that heads toward the dipole magnetic field, and in the tailward outflow that has no obstacle ahead (e.g. Runov et al., 2015, 2018). Thus, a realistic model with the dipole field at the earthward side of the simulation is needed to accurately describe the ion kinetics within the plasma sheet.

Three-dimensional global hybrid simulations of storm-time magnetospheric dynamics with a focus on the magnetotail were performed in Lin et al. (2014). The model reproduced well the magnetotail dynamic features such as bi-directional fast flows





generated by reconnection, transient magnetic field structures like DFBs and flux ropes, particle energization and injection into the inner magnetosphere. The simulated ion VDFs earthward ($X = -16\,R_{\mathrm{E}}$) and tailward ($X = -31\,R_{\mathrm{E}}$) of the reconnection site were shown to be multicomponent with lower energy inflow and higher energy outflow populations. Closer to the dipole ($X = -10\,R_{\mathrm{E}}$), where azimuthal drifts prevail over earthward convection, the ion VDFs were fairly gyrotropic and showed

significant heating in the direction perpendicular to the magnetic field, indicating the betatron energization. The results showed many similarities with the observations.

It is worth noting, however, that particle kinetics during magnetic reconnection is extremely complex. It involves many processes, such as local particle acceleration by Fermi and betatron processes, pick up of non-local particles, Speiser-type particle motion (Speiser, 1965) and energization in the reconnection electric field (e.g., Birn et al., 2012). Modern simulations

are numerical experiments and require analysis similar to experimental data analysis to understand simulation results. However, contrary to observations, simulations have an advantage of making different simplifications. Specifically, for the problem of ion kinetics in reconnection and reconnection-related processes, the first step may be done with utilization of two-dimensional (2D) models. In 2D some aspects of ion transport, like replenishment of plasma evacuated from the reconnection site from the ambient plasma sheet, are not present. That allows to study the dynamics in a simpler setup. After 2D effects are understood,

the results of a numerical experiment with 3D dynamics will be easier to comprehend. Moreover, the global hybrid models that describe ion kinetics by solving the equation of particle motion (Lin et al., 2014), although they provide a great tool to study ion scale dynamics, often produce noisy distributions due to insufficient numbers of particles. Models that solve the Vlasov equation for distribution functions provide noiseless VDFs and, therefore, are a valuable tool to study the ion VDF evolution associated with magnetotail dynamics. The fundamental question to be addressed with this type of models is the evolution of

ion VDFs during dynamic processes such as magnetic reconnection.

In this paper we discuss the results of 2D global hybrid-Vlasov modeling of magnetotail reconnection with the Vlasiator model (Palmroth et al., 2018). We use a run including the entire magnetosphere in 2D. The tail dynamics of this run has been discussed in a number of papers (e.g., Palmroth et al., 2017; Juusola et al., 2018a, b; Grandin et al., 2019b). Most recently, Grandin et al. (2019b) found that in this simulation, the transition region between the tail-like and dipolar field acts

like a buffer and permits only some particles to precipitate into the ionosphere. Here, we aim to understand the tail VDFs in terms of reconnection processes in a self-consistent ion-kinetic setup using such a large simulation domain that it covers the reconnection regions, the outflows, and also includes a realistically sized Earth's dipole. The simulation results are compared with *in situ* observations by THEMIS and ARTEMIS spacecraft in the magnetotail. The paper is organized as follows: First, in Section 2, we introduce briefly the Vlasiator model and THEMIS and ARTEMIS missions. In Section 3, we introduce the

VDF characteristics within the earthward and tailward outflow regions and perform model-data comparisons. In Section 4, we end the paper with a discussion.



## 2 Model and data

### 2.1 Global hybrid-Vlasov simulation with Vlasiator

Vlasiator is a global hybrid-Vlasov model of the Earth's magnetosphere, which describes protons in the six-dimensional (6D) phase space (von Alfthan et al., 2014). A recent review (Palmroth et al., 2018) describes the set of equations along with other code design features. Vlasiator describes ion physics self-consistently at ion-kinetic scales, featuring proton velocity distribution functions throughout the simulation box without statistical sampling noise. Electrons are a massless charge-neutralizing fluid, indicating that it is assumed that the electron pressure influence on Ohm's law is negligible. Unique in Vlasiator compared to many other simulations in the kinetic global regime is that the Earth's dipole strength represents the actual conditions within near-Earth space in 2D spatial setups (see Daldorff et al., 2014). Therefore, the resulting temporal and spatial scales in the simulation can be directly compared to the Earth's magnetosphere without scaling. The temporal and spatial scales are given in SI units to facilitate direct comparisons.

An overview of the simulation run investigated in this paper is given in Figure 1 (see also Supplementary movie S1 in Palmroth et al. (2017)), color-coding proton number density in the simulation domain. While Vlasiator is fully 6D, here we outline a 5D run, representing 3D velocity distributions in the 2D noon–midnight polar plane (2D3V description). The simulation takes place in the geocentric solar ecliptic (GSE) (X,Z) plane, in which the solar wind is therefore incoming from the right-hand-side wall of the simulation box, located at $X = 48\,R_{\mathrm{E}}$. The left-hand-side wall at $X = -94\,R_{\mathrm{E}}$ defines the boundary of the simulation domain in the nightside, and the extent of the simulation box in the Z direction is comprised between the walls at $Z = \pm\,56\,R_{\mathrm{E}}$. The boundary conditions for these walls (except the one from which the solar wind is flowing in) are Neumann conditions, in which the normal derivative of each parameter is set to zero by copying values to the neighboring ghost cells. Periodic boundary conditions are applied in the out-of-plane direction ($Y$ axis in the GSE frame). The solar wind bulk parameters are as follows: density of $1\,\mathrm{cm}^{-3}$, velocity of $750\,\mathrm{km\,s}^{-1}$ in the –X direction, and the IMF is purely southward with a magnitude of 5 nT. Such solar wind conditions are representative of that which can be observed during solar wind high-speed stream events (e.g., Grandin et al., 2019a). The solar wind ion population is initialized with a uniform Maxwellian velocity distribution function, and the whole magnetosphere forms self-consistently, driven by the constant solar wind inflow and the 2D dipole field. The inner boundary of the simulation box is a perfectly conducting sphere located at $4.7\,R_{\mathrm{E}}$ from the Earth's center. In this run, the ordinary space has a resolution of $300\,\mathrm{km}$ and the velocity space has a resolution of $30\,\mathrm{km\,s}^{-1}$, in the whole domain. This choice makes it possible to investigate ion-scale physics self-consistently.

### 2.2 THEMIS and ARTEMIS

We use observations from Time History of Events and Macroscale Interactions during Substorms (THEMIS, Angelopoulos, 2008) and Acceleration, Reconnection, Turbulence and Electrodynamics of Moon's Interaction with the Sun (ARTEMIS, Angelopoulos, 2011) multi-probe spacecraft to compare them with the Vlasiator simulation results. The THEMIS and ARTEMIS probes are identical and equipped with the Flux Gate Magnetometer (FGM, Auster et al., 2008), the Electric Field Instrument (EFI, Bonnell et al., 2008), the Electrostatic Analyzer, (ESA, McFadden et al., 2008) and the Solid State Telescope (SST,

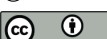

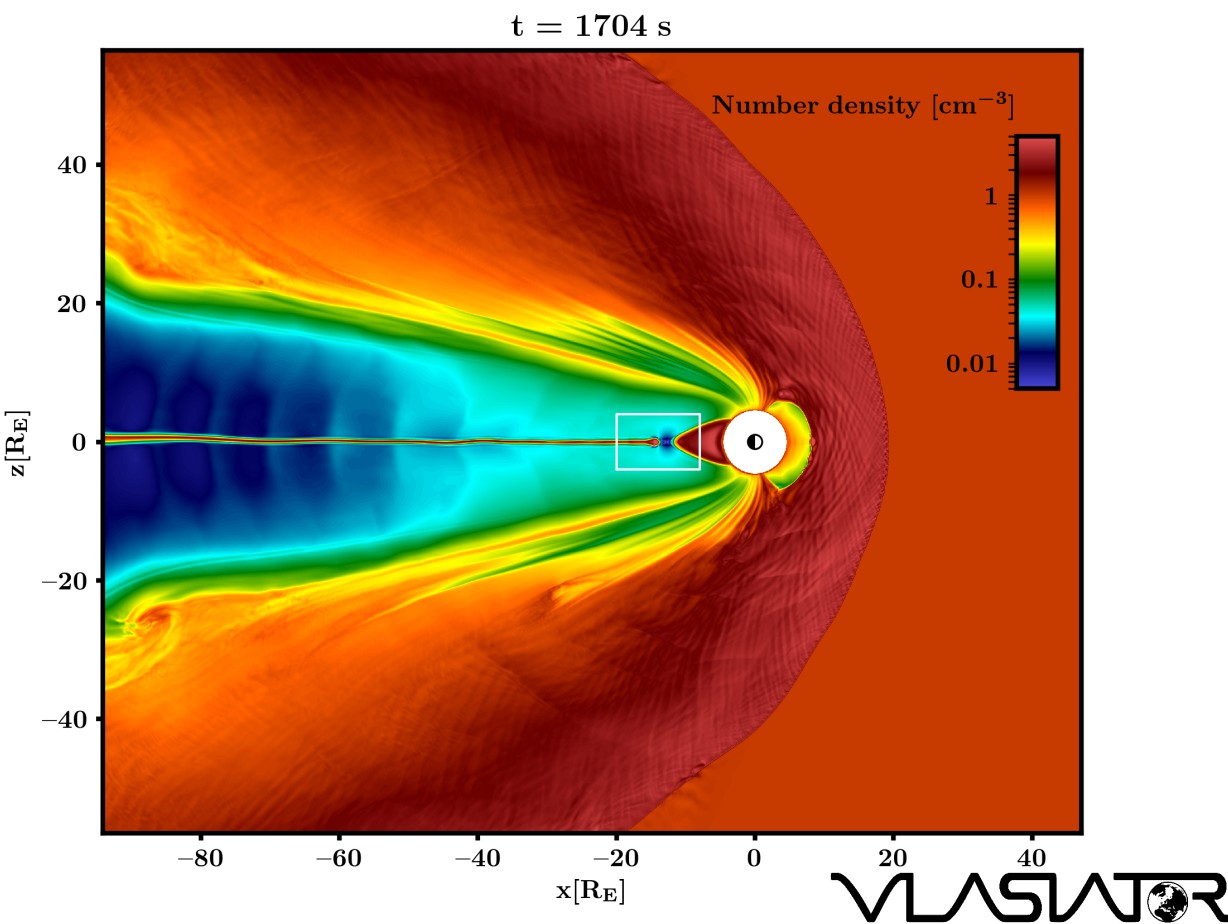

**Figure 1.** Overview of the Vlasiator simulation run investigated in this paper. Color-coding depicts the proton number density at time $t = 1704$ s from the beginning of the simulation. The plasma sheet shows that reconnection has begun shortly before the depicted frame. The white rectangle shows the area on which the subsequent figures will focus.





Angelopoulos et al., 2008b). In the 2008–2009 flight years, the five THEMIS probes were on equatorial orbits and formed the Sun–Earth elongated conjunction at distances $7 < R < 30\,R_{\mathrm{E}}$ during the magnetotail science seasons. In 2010, the two most-distant probes were injected to Lunar orbit and traversed the Earth magnetotail with the Moon at $R \sim 60\,R_{\mathrm{E}}$ during about 3 days per month. For the purposes of this study, we use the ion distribution function calculated from particle fluxes collected

by ESA in the energy range from $\sim10\,\mathrm{eV}$ to $30\,\mathrm{keV}$ and by SST in the energy range from $\sim50\,\mathrm{keV}$ to $300\,\mathrm{keV}$. The instrument response simulations with the GEANT-4 model were used to fill the gap between ESA and SST energy ranges (Runov et al., 2015).

## 3    Simulation Results and the Model-data Comparison

In the following subsections, we present a comprehensive analysis of the obtained ion velocity distribution functions (VDFs)

within the tail. To illustrate our purpose, we show in Figure 2 an overview of the tail area and the analysis methods. The time instant in Figure 2 is the same as in Figure 1, i.e., $t = 1704\,\mathrm{s}$. Figure 2 illustrates the reconnection points in the tail, marked by red crosses, computed by identifying the saddle points in the magnetic flux function. The method is outlined in Hoilijoki et al. (2017) and further developed in Hoilijoki et al. (2019).

As time progresses, the main reconnection line has developed near $X \approx -13\,R_{\mathrm{E}}$. The advantage of global hybrid simula-

tions is that they allow us to study self-consistently evolving ion VDFs at a variety of spatial and temporal locations. Moreover, the VDFs simulated with the hybrid-Vlasov technique do not suffer from sampling noise, and provide distributions without temporal or spatial integration, which helps Vlasiator excel at VDF analysis of moving structures and comparison with observations. Snapshots of ion VDF 2D cuts in the $\{v_x, v_z\}$ plane at $Z = 0$ earthward and tailward of the main X-line are shown in Figure 2b–f. The VDFs are taken at positions marked with circles of matching colors in Figure 2a, and they are shown in the

plasma frame (i.e., $\mathbf{v} = 0$ corresponds to the local bulk velocity $\mathbf{V}$). The black arrows depict the magnetic field direction and relative strength at the center of the colored circles. Figure 2 shows that the cold and isotropic ion VDFs flowing in from the lobes have become structured while reconnection has developed.

### 3.1    Earthward Outflow

Figure 3 shows ion VDFs in the Vlasiator simulation detected earthward of the major X-line at three time instances. At

$t = 1662.0\,\mathrm{s}$ (simulation time) reconnection at $X \approx -13\,R_{\mathrm{E}}$ starts to develop (Fig. 3a). The unbalanced magnetic tensions on the reconnected field lines start to force the stretched field lines at $X > -13\,R_{\mathrm{E}}$ to dipolarize. Slices of the VDF at the location indicated with a cyan circle are shown in Fig. 3b–d. The cuts in velocity space are made in the $\{v_{\parallel}, v_{\perp_1}\}$, $\{v_{\parallel}, v_{\perp_2}\}$, and $\{v_{\perp_1}, v_{\perp_2}\}$ planes, respectively. The $v_{\parallel}$ direction is defined along the local magnetic field $\mathbf{B}$, $v_{\perp_1}$ is defined along the $\mathbf{B} \times \mathbf{V}$ direction, and $v_{\perp_2}$ is defined along the $\mathbf{B} \times (\mathbf{B} \times \mathbf{V})$ direction; the thus defined velocity frame is depicted in Fig. 3a. At

$t = 1662.0\,\mathrm{s}$, the initially isotropic ion VDF at $X \approx -12\,R_{\mathrm{E}}$ starts developing transverse anisotropy (Fig. 3b–c). The $\{v_{\perp_1}, v_{\perp_2}\}$ cut (Fig. 3d) shows that the ion VDF remains gyrotropic.

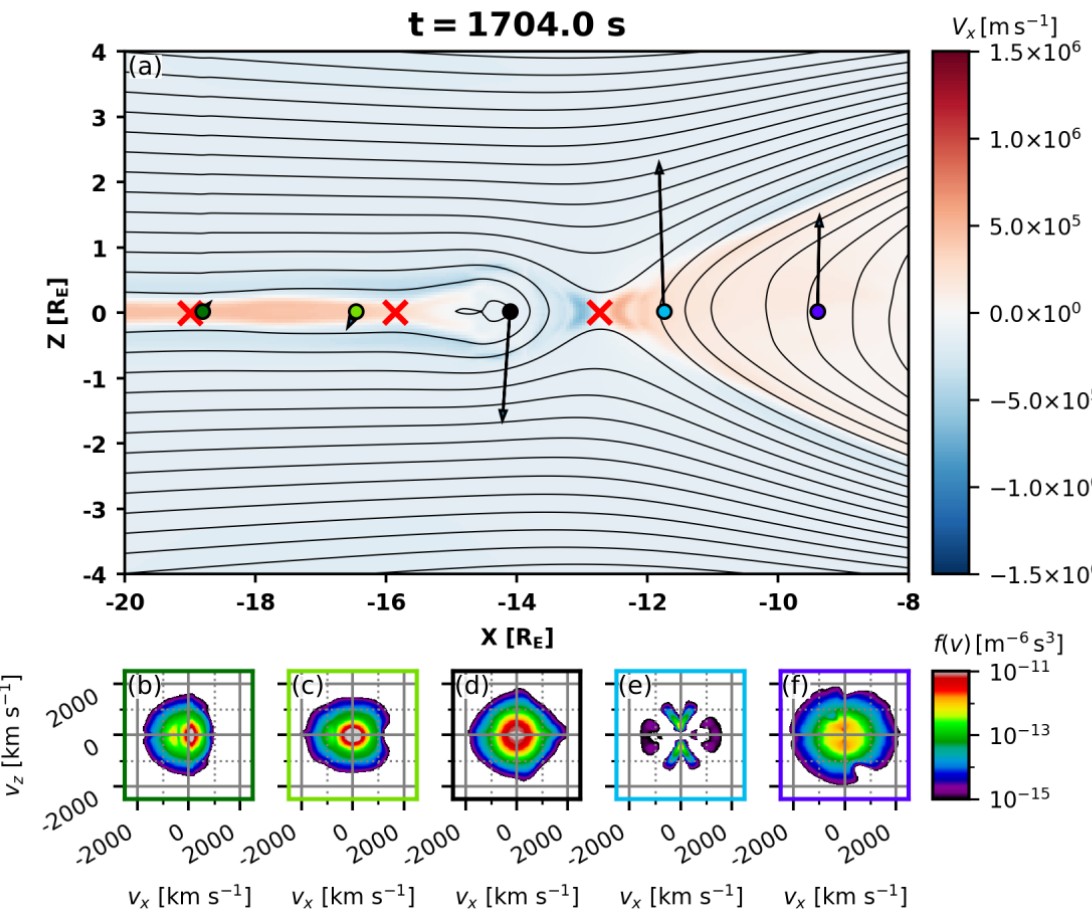

**Figure 2.** (a) X-component of the plasma bulk velocity in the magnetotail at $t = 1704.0$ s, in the area depicted with a white rectangle in Fig. 1. Blue (red) colors illustrate tailward (earthward) plasma flows. The black contours show magnetic field lines, and red crosses indicate X-points. The black arrows show the direction and relative magnitude of the magnetic field projected into the XZ plane at the five locations indicated with colored circles. (b–f) Slices of the 3D VDFs in the $\{v_x, v_z\}$ plane at those five locations. The VDFs are shown in the plasma frame. VDF: Velocity distribution function.





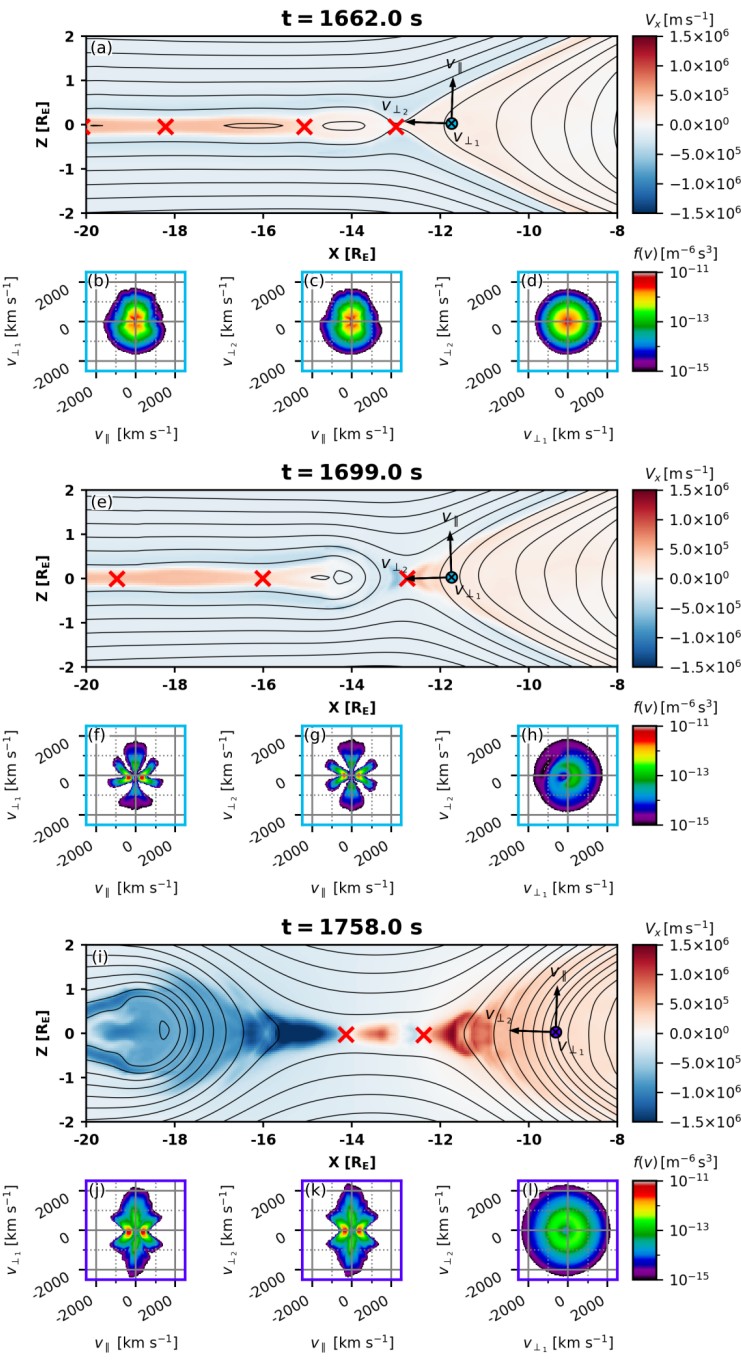

**Figure 3.** (a) X-component of the plasma bulk velocity in the magnetotail at $t = 1662.0$ s, with magnetic field lines and X-points indicated as in Fig. 2. A virtual spacecraft is placed near ($X = -12\,R_{\mathrm{E}}$, $Z = 0$) to monitor the plasma earthward from the major X-line; arrows and labels indicate the local plasma frame (see text for details). (b–d) Slices of the 3D VDF at the virtual spacecraft in the $\{v_{\parallel}, v_{\perp_1}\}$, $\{v_{\parallel}, v_{\perp_2}\}$, and $\{v_{\perp_1}, v_{\perp_2}\}$ planes. (e–h) Same at $t = 1699.0$ s. (i–l) Same at $t = 1758.0$ s but using a different virtual spacecraft, observing plasma on the same magnetic field line as in (e–h) after it has convected earthward. VDF: Velocity distribution function.





At $t = 1699.0$ s, reconnection develops and the earthward bulk velocity increases (Fig. 3e). The ion VDF at $X \approx -12\,R_{\mathrm{E}}$ becomes structured developing transversal anisotropy at high energies. Fig. 3f–g show two parallel counter-streaming beams appearing at lower energies. Phase-space density lacunae (i.e., empty regions) appear at pitch angles of $\pm 10°$ and $\pm 45°$ in the $v_{\parallel} > 0$ and $v_{\parallel} < 0$ half-spaces. The phase-space density gaps at low pitch angles are consistent with observations (Runov et al.,

2017) and test-particle simulations (Birn and Runov, 2016), and may be explained by the loss of particles along the field lines (Runov et al., 2017). The gaps at $\sim 45°$ would require particle tracing to be explained. It is worth noting that at $t = 1699.0$ s the virtual probe records a lower proton density than at $t = 1662.0$ s, since the outflow jets evacuate the plasma sheet population and the only source of plasma replenishment is the inflow from the low-density lobes. In real 3D scenarios, the situation is more complex because the flank plasma sheet also contributes to refilling, a contribution that is variable depending on the spatial

extent of the X-line. The cold lobe plasma has been energized by two processes: betatron acceleration, which manifests in the transverse ion VDF anisotropy, and Fermi acceleration, which produces the parallel beams. Fig. 3h shows that the ion VDF becomes agyroptropic with higher phase-space density in the $v_{\perp_1} > 0$ half-space (note that the bulk velocity is subtracted from the VDF).

At $t = 1758.0$ s, an O-point forms at $X \approx -13.0\,R_{\mathrm{E}}$ between two X-lines; the magnetic field at $X > -11\,R_{\mathrm{E}}$ is compressed

by earthward outflow from the main X-line which has migrated to $X \approx 12.5\,R_{\mathrm{E}}$ (Fig. 3i). At this time, we place our virtual detector closer to the Earth at $X \approx -9.5\,R_{\mathrm{E}}$, on the same magnetic flux tube which was studied at $X \approx -12\,R_{\mathrm{E}}$ at $t = 1699.0$ s in Fig. 3e–h. The $\{v_{\parallel}, v_{\perp_1}\}$ and $\{v_{\parallel}, v_{\perp_2}\}$ ion VDF cuts (Fig. 3j–k) show the same structure as at $t = 1699.0$ s: transverse anisotropy at high energies and counter-streaming parallel beams at lower energies. Note that the lacunae at $\sim 45°$ pitch-angles are now partly filled, whereas lacunae at low pitch angles persist. Contrary to that at $t = 1699.0$ s, the $\{v_{\perp_1}, v_{\perp_2}\}$ cut (Fig. 3l)

shows a rather gyrotropic ring-type distribution.

## 3.2  Tailward Outflow

Figure 4 shows magnetic field lines, X-component of the plasma velocity, and ion VDF cuts tailward of the reconnection region at three time instances in a similar setup as Figure 3. Now our aim is to follow the evolution of ion distributions during reconnection development from an early stage at $t = 1742.0$ s simulation time to a more mature state at $t = 1747.0$ s and to

investigate O-line formation at $t = 1766.0$ s.

At $t = 1742.0$ s, when tailward reconnection outflow from the X-line at $X \approx -13.3\,R_{\mathrm{E}}$ is about to reach the virtual detector location ($X \approx -16.5\,R_{\mathrm{E}}$, Fig. 4a), the ion VDF in the plasma velocity frame is quasi-Maxwellian without considerable structuring (Fig. 4b–d). The situation changes dramatically when the front of anti-dipolarization (increased negative $B_z$ embedded into the tailward outflow) reaches the virtual detector at $t = 1747.0$ s (Fig. 4e). The ion distribution cuts in the flux tubes with

increased $|B_z|$, shown in Fig. 4f–h, exhibit distinct features: a strong transverse anisotropy at high energies, counter-streaming parallel beams at lower energies, and a pronounced agyrotropy. It is worth noting that the observed changes in the ion VDFs are due to arriving of freshly reconnected plasma population within reconnected flux tubes, i.e., they characterise a spatial effect rather than temporal evolution of the VDF observed earlier. To follow the temporal evolution of the ion VDF, at $t = 1766.0$ s we move our virtual detector tailward to $X \approx -19\,R_{\mathrm{E}}$ along with the reconnected flux tubes (Fig. 4i). The ion VDF at this





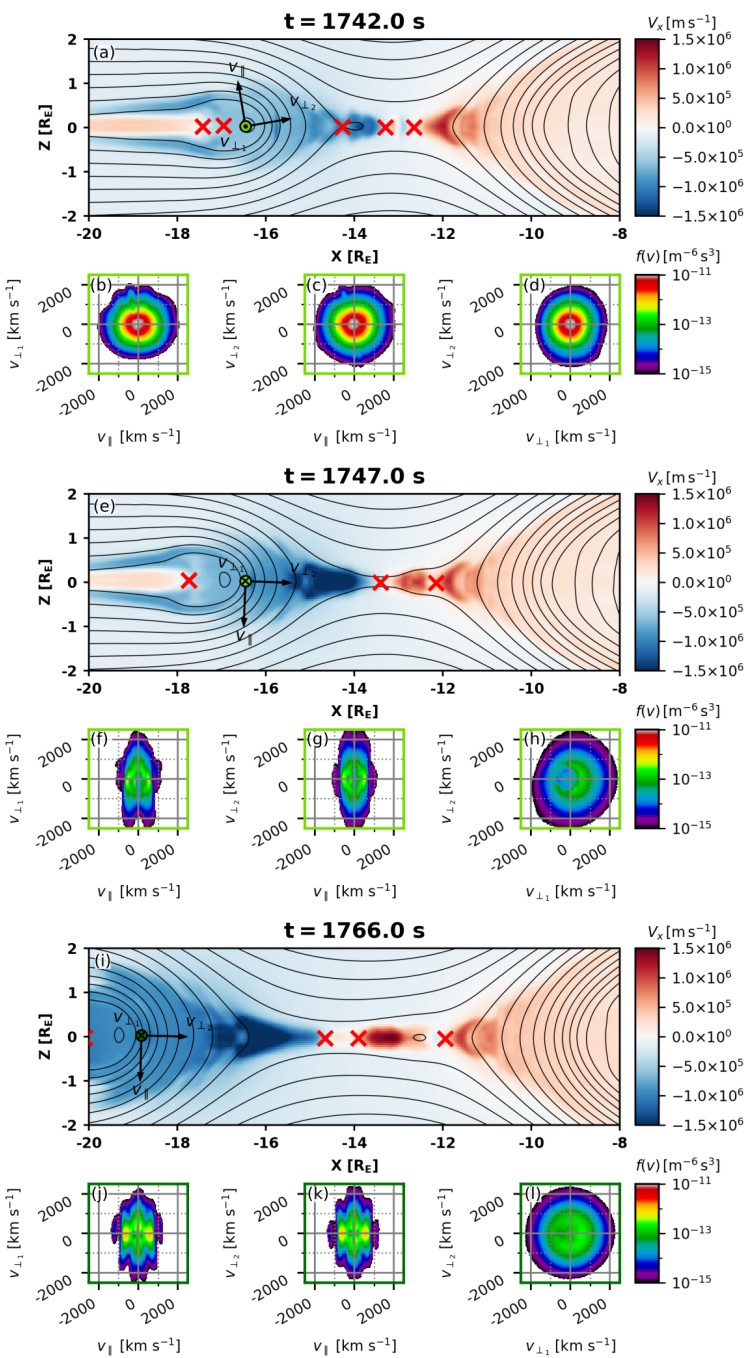

**Figure 4.** Same format as in Fig. 3 but focusing on the plasma tailward from the major X-line, at (a–d) $t = 1742.0$ s, (e–h) $t = 1747.0$ s, and (i–l) $t = 1766.0$ s. Note that the plasma monitored in (i–l) is located on the same magnetic field line as in (e–h) after it has convected tailward.

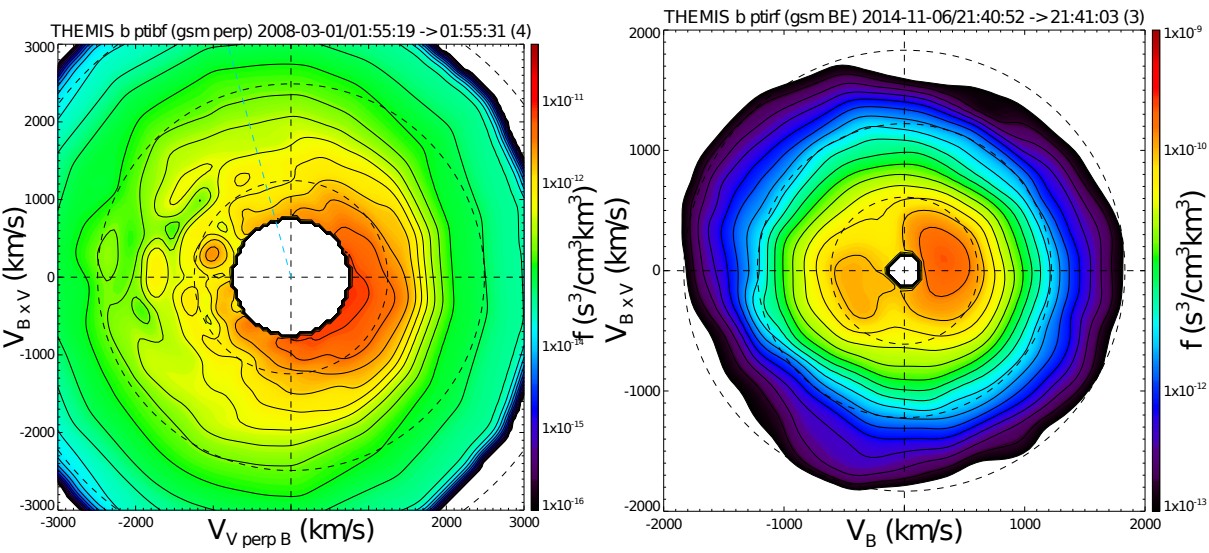

**Figure 5.** Examples of the ion velocity distribution functions observed within tailward reconnection outflows by THEMIS at $X \sim -23\,R_{\mathrm{E}}$ (left) and by ARTEMIS at $X \sim -60\,R_{\mathrm{E}}$ (right). The distributions are shown in the bulk velocity frame. Dashed circles indicate $[1, 2, 3] \times V_{th}$, where $V_{th}$ is the ion thermal velocity.

spatio-temporal location shows, generally, same features as at the earlier stage at $X \approx -16.5\,R_{\mathrm{E}}$, but the parallel beams have shifted toward higher energies (Fig. 4j–k) and the agyrotropy has become less pronounced (Fig. 4l). The shift of the parallel beams toward higher energies indicates Fermi acceleration.

Figure 5 shows two examples of ion VDFs observed within tailward reconnection outflows at $X \sim -23\,R_{\mathrm{E}}$ and $X \sim -60\,R_{\mathrm{E}}$
5  by THEMIS-B on 1 March 2008 at 01:55 UT and ARTEMIS on 6 November 2014 at 21:41 UT, respectively. Both probes detected an enhancement in the magnetic flux transport rate $[\mathbf{V} \times \mathbf{B}]_y > 2\,\mathrm{mV/m}$, which allows us to interpret the observed fast flows as reconnection ejecta (Runov et al., 2018). The VDFs are collected in the frame of reference moving with the instantaneous bulk velocity. The $\{\mathbf{v}_\perp, \mathbf{v}_{\mathbf{V} \times \mathbf{B}}\}$ (where $\mathbf{v}_\perp = (\mathbf{B} \times [\mathbf{V} \times \mathbf{B}])/B$) plane cut of the VDF observed at $X \sim -23\,R_{\mathrm{E}}$ (left panel) indicates a pronounced agyrotopy of the thermal ion population. This resembles the agyrotropic VDF that appears
10  at $t = 1747.0\,\mathrm{s}$ (Figure 4h). The ion VDF observed by ARTEMIS (right panel) is characterized by counter-streaming field-aligned beams of subthermal and nearly thermal ions. The low-energy counter-streaming beams appear in simulated VDFs at $t = 1747.0\,\mathrm{s}$ and $t = 1766.0\,\mathrm{s}$ (Figure 4f and 4j).

## 3.3 Quasi-steady X-line

Figure 6a shows the magnetic field configuration (contours) and the ion bulk velocity X-component (colors) in the Vlasiator
15  simulation at $t = 1950.5\,\mathrm{s}$ at and around a quasi-steady X-line near $X \approx -14\,R_{\mathrm{E}}$. Because the magnetic field is very weak at and near the X-line, to avoid the uncertainty associated with the mapping to the local field-aligned coordinate system, we use





**Figure 6.** (a) X-component of the plasma bulk velocity in the magnetotail at $t = 1950.5$ s, in the same format as in Fig. 2, with magnetic field lines, X-point locations, five virtual spacecraft and the magnetic field projection in the XZ plane at their locations. Slices of the 3D VDFs at the five locations indicated with colored circles are shown in the (b–f) $\{v_x, v_z\}$ plane, (g–k) $\{v_x, v_y\}$ plane, and (l–p) $\{v_y, v_z\}$ plane. The VDFs are shown in the plasma frame. VDF: Velocity distribution function.

simulation coordinates (GSE) and show cuts of the 3D ion VDFs in the $\{v_x, v_z\}$ (Fig. 6b–f), $\{v_x, v_y\}$ (Fig. 6g–k), and $\{v_y, v_z\}$ (Fig. 6l–p) planes at the five virtual detectors of matching colors shown in Fig. 6a.





At the time depicted in Figure 6, the simulation exhibits a main X-line at $X \approx -14\,R_{\mathrm{E}}$ and a secondary one at $X \approx -12\,R_{\mathrm{E}}$, indicated by the red crosses. The position of the virtual detector placed near the main X-line is shown by the black circle in Fig. 6a. The magnetic field at this position is weak and directed mainly along $X$. The corresponding ion VDF cuts, shown in the black boxes, indicate a half-disc, strongly agyrotropic type of distribution. The $\{v_y, v_z\}$ cut (Fig. 6n), nearly perpendicular to the instantaneous magnetic field, indicates acceleration in the positive $Y$ direction, i.e., along the cross-tail and reconnection electric fields. The high phase-space density population between $-1000 < v_y < 0\,\mathrm{km\,s^{-1}}$ and $-1000 < v_z < 0\,\mathrm{km\,s^{-1}}$ is the inflowing low-energy ion population. It is interesting that the entire distribution is shifted in the $\{v_y, v_z\}$ space toward negative $v_y$, which is likely a signature of the Hall electric field $E_z$, normal to the current sheet (e.g., Hesse et al., 1998). The presence of the Hall electric field in a thin current sheet prior to and during reconnection has also been shown in a number of observations (see, e.g., Lu et al., 2019, and references therein).

The ion velocity distribution obtained by the virtual detector placed near the secondary X-line at $X \approx -12\,R_{\mathrm{E}}$ (light blue circle in Fig. 6a and VDF slices in Fig. 6e,j,o) is also agyrotropic, showing acceleration by the reconnection and cross-tail electric fields along positive $Y$. Unlike the strict electric field acceleration along $Y$ near the main X-line, ions near the secondary X-line are also accelerated in the positive $X$ direction (i.e., earthward). This is caused by the gyration in the increasing northward $B_z$. Similar distributions are detected in the tailward outflow at $X \approx -16.5$ (light green circle in Fig. 6a and VDF slices in Fig. 6c,h,m). In this case, ions are accelerated in the $+Y$ direction by the reconnection and cross-tail electric fields and in the $-X$ direction (tailward) due to gyration in the increasing negative $B_z$. Notably, hook-like distributions in the $\{v_x, v_y\}$ plane in earthward (at $X \approx -12\,R_{\mathrm{E}}$) and tailward ($X \approx -16.5$) outflows are nearly mirror-symmetric. The described VDF signatures indicate a Speiser-type (Speiser, 1965) ion motion earthward and tailward of the reconnection line.

It is worth mentioning the dotted structure of the Speiser hook-like distributions. Very similar structures were also reported in Nakamura et al. (1998) and interpreted as gyro-bunch of cold ions. In the Vlasiator run this effect is caused by cold ions that came in from the lobes. This effect, most likely, is not observable in situ at near-Earth reconnection places because of the 3D nature of reconnection there that causes hot plasma sheet ion inflow from the dawn side (e.g., Birn et al., 2015). At lunar distances, however, this effect may be observed by instruments with sufficient angular and time resolution. Signatures of the cold ions gyro-bunching were also noted in 2.5D PIC simulations of lunar-distant tail reconnection (Hietala et al., 2015). The twist (or rotation) of the hook-like distribution is due to the Hall magnetic field, which twists the reconnected field lines. This effect was also reported in PIC simulations (Hietala et al., 2015).

Figure 6f,k,p show the ion VDF detected at $X \approx -9.5\,R_{\mathrm{E}}$ (dark violet circle in Fig. 6a), where the magnetic configuration sharply changes from stretched to more dipolar field lines. The magnetic field lies mainly in the XZ plane, thus the VDF $\{v_x, v_z\}$ cut (Fig. 6f) is approximately in the $\{v_\parallel, v_{\perp_2}\}$ plane. Accelerations in the $\pm v_z$ directions and in the positive $v_x$ direction are visible. They depict the operation of Fermi-type field-aligned ion acceleration due to flux tube shrinking. The VDF $\{v_x, v_y\}$ (Fig. 6k) and $\{v_y, v_z\}$ (Fig. 6p) cuts indicate ion motion in the positive $v_y$ direction, which is likely due to the gradient drift of the ions energized by the convective electric field $E_y \propto V_x B_z$. This effect was reported in observations (Runov et al., 2015, 2017) and in test-particle modeling (Birn et al., 2015; Birn and Runov, 2016).





The ion VDF obtained near the O-line at $X \approx -19\,R_{\mathrm{E}}$ (dark green circle in Fig. 6a) is shown in Fig. 6b,g,l. At this location, the magnetic field is directed predominantly along $-Z$. The VDF shows a shell-like structure. It is generally gyrotropic (see the $\{v_x, v_y\}$ cut, Fig. 6g), with distinct counter-steaming field-aligned beams which can be seen in the $\{v_x, v_z\}$ (Fig. 6b) and $\{v_y, v_z\}$ cuts (Fig. 6l).

## 4 Discussion and Concluding Remarks

We present results of global hybrid-Vlasov simulations of magnetotail dynamics during southward IMF (see Palmroth et al., 2017; Juusola et al., 2018a, for simulation details). The simulation results include 2.5D magnetic field, plasma moments, and 3D velocity distribution functions (VDFs). The simulation shows that reconnection with multiple X-points occurs in the magnetotail with the main X-point near $X \approx -13\,R_{\mathrm{E}}$. We have examined the VDFs that were observed by virtual detectors placed in the vicinity of the main X-point and within earthward and tailward reconnection outflows. The simulation setup (spatial resolution of 300 km) allows the study of this region, as the ion inertial length $d_i$ is well resolved at the locations of the virtual detectors: from the leftmost (dark green) to the rightmost (dark violet) virtual detector in Fig. 6a, $d_i$ is 433, 580, 929, 669 and 571 km.

The global hybrid-Vlasov simulations allow us to place virtual detectors at a variety of locations following the reconnected flux tubes along their transport toward the dipole within the earthward outflow and toward the weaker magnetic field within the tailward outflow. Vlasiator successfully reproduces characteristics of VDFs that were previously reported in observations: the transverse anisotropy of the energetic ion population that is associated with the betatron ion energization and counter-streaming ion beams at lower energies, which are signatures of Fermi-type ion acceleration along the contracting magnetic flux tubes. These effects have been observed in the near-Earth plasma sheet during substorm dipolarizations (Delcourt et al., 1997) and within dipolarizing flux bundles embedded into earthward flows (Runov et al., 2017) and appeared in particle-in-cell and test-particle simulations (Pritchett and Runov, 2017; Birn and Runov, 2016). The Vlasiator simulation allows us in addition to study the ion distribution at a later stage, after the magnetic field line on which the plasma was located has convected earthward. It is found that the distribution becomes more gyrotropic but retains the transverse anisotropy and the counter-streaming ion beams when being convected earthward. Lacunae observed near the reconnection X-point are on the other hand partly filled by diffusion processes when reaching the most earthward virtual detector.

The virtual detectors placed along the tail within the tailward outflow also present some signatures that have been detected in-situ. Specifically, agyrotropy of energetic ions is observed, appearing in the simulations at the strong $B_z$ gradient at the earthward edge of the plasmoid (or anti-dipolarization front, see, e.g., Li et al., 2014). We find a similar feature in the VDFs observed tailward of a reconnection X-line. Another distinct characteristic that appears in simulated and observed ion VDFs is the presence of the counter-streaming field-aligned ion beams at lower energies. However, some of the ion VDF characteristics, such as the pronounced transversal anisotropy of energetic ions within tailward reconnection outflow, are not present in the VDFs collected in-situ. This may be because THEMIS probes' ability to measure ion pitch-angle distributions at energies higher than 30 keV is somewhat limited. Again, the simulation allows us to study the evolution of the ion distribution at a later





stage and further tailward, finding that the features are overall preserved but that the parallel beams are seen at higher energies, indicating that Fermi acceleration continues during the tailward convection of the plasma. In addition, the agyrotropy observed near the X-point is less prominent at the most tailward location.

After magnetic flux and plasma were evacuated earthward and tailward from the vicinity of the main reconnection line, a

quasi-steady reconnection with secondary X-lines developed at $X \approx -14\,R_{\mathrm{E}}$. The ion VDFs at virtual detectors placed along the reconnecting current sheet reveal the presence of ion populations accelerated duskward by the reconnecting electric field near the X-line and ions performing Speiser-type motion in reconnecting magnetic field earthward and tailward from the X-line. It is very difficult to observe, and therefore study, such distributions in situ in the near-Earth magnetotail, where the X-line cross-tail length is limited (e.g., Nagai et al., 2013). Because of this 3D nature, the plasma evacuated with reconnected flux

tubes is replenished by the ambient plasma sheet particles (Runov et al., 2015; Birn et al., 2015). At lunar distances, where the ambient plasma sheet ion population is cooler, Speiser-type ion distributions within the reconnection jet, similar to that shown in Figure 6, have been observed (Hietala et al., 2015). While inherently a limitation of the simulation setup, the 2D nature of the Vlasiator run proves useful to specifically study the ion distributions associated with Speiser-type motion in the vicinity of a quasi-stable X-line.

To conclude, we find that the Vlasiator VDFs reasonably represent the observed VDFs during similar conditions reported in the previous literature using in situ satellite observations, indicating that the 2D hybrid-Vlasov simulations provide a valuable tool to study the evolution of ion velocity distribution functions. The simulated distribution functions may be interpreted in terms of basic particle acceleration mechanisms, such as the Fermi-type and betatron accelerations, the direct acceleration of unmagnetized particles by the reconnection electric field, and the Speiser-type ion motion. Some important 3D effects, such

as Shabansky-type ion acceleration by the moving dipolarization front (Zhou et al., 2010), however, are not present due to the 2D model setup limitations. Another valuable tool that needs to be included into the modeling is backward particle tracing in time using Liouville's theorem. This method helps to understand how the particle distributions were formed (e.g., Birn et al., 2015). This instrument is under development. It is worth stressing again that 2D global hybrid simulations with the Vlasiator code provide very helpful insight into the formation and evolution of ion distribution functions, allowing us in particular to

analyse how a given ion distribution evolves as the magnetic field line to which it is associated is convected away from the reconnection site (earthward or tailward). Comparisons with in situ observations by the THEMIS and ARTEMIS probes have shown remarkable similarity in ion VDFs observed within reconnection outflows in the magnetotail and those resulting from Vlasiator simulations. Yet some features of ion VDFs discovered in the discussed Vlasiator run are not readily observable in situ because of the 3D nature of near-Earth reconnection and/or due to instrumental limitations. The VDFs categorized in this

paper may however be used as predictions for future magnetotail observations with higher temporal and angular resolutions than available now.

*Acknowledgements.* This paper was outlined and drafted during the Second International Vlasiator Science Hackathon held in Helsinki, 20-24 August, 2018. The Hackathon was funded by the European Research Council grant 682068-PRESTISSIMO. We acknowledge the



European Research Council for Starting grant 200141-QuESpace, with which Vlasiator (`http://helsinki.fi/vlasiator`) was developed, and Consolidator grant 682068-PRESTISSIMO awarded to further develop Vlasiator and use it for scientific investigations. We gratefully also acknowledge the Academy of Finland (grant number 309937). The Finnish Centre of Excellence in Research of Sustainable Space, funded through the Academy of Finland with grant number 312351, supports Vlasiator development and science as well.

5 The run shown in this paper was carried out with a Tier-0 PRACE grant, project number 2014112573, on HazelHen/HLRS. LT is supported by Academy of Finland grant agreement No 322544. HH's work was supported by Royal Society University Research Fellowship URF\R1\180671 and the Turku Collegium for Science and Medicine. We acknowledge NASA contract NAS5-02099 and V. Angelopoulos for use of data from THEMIS and ARTEMIS Missions. THEMIS and ARTEMIS data are available on http://themis.ssl.berkeley.edu/ and via SPEDAS system https://spedas.org/. We thank Thiago Brito for help with software and Vlasiator data analysis.



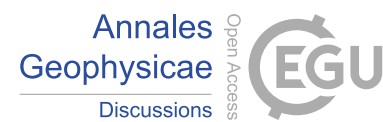

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
