# Peer review of "Ion distribution functions in magnetotail reconnection: Global hybrid-Vlasov simulation results"

_Annales Geophysicae, 2020_

## Author Response (AR1)

**Manuscript angeo-2020-89: "Ion distribution functions in magnetotail reconnection: Global hybrid-Vlasov simulation results"**
**by A. Runov et al.**

We thank the two Reviewers for their careful evaluation of our manuscript and for the valuable suggestions. Our responses to the comments are shown in blue, while the original text written by the Reviewers is shown in black. When we refer to line numbers, it should be understood in the revised manuscript with highlighted changes.

**Response to Reviewer #1**

The authors carried out a comprehensive analysis on Vlasiator simulation results of magnetotail reconnection, focusing especially on the ion distribution functions and their comparison with spacecraft observations. Although we cannot expect a full consistency between the simulated and observed ion distributions (for example, the phase-space lacunae at the pitch angle of 45 degrees is indeed very puzzling), the simulation results indeed show some of the observational features, which in some sense validates the Vlasiator model and provides a useful tool to understand the particle dynamics associated with the reconnection process. I believe that the paper is overall a good addition to our current knowledge of magnetic reconnection, although I have some minor comments listed below.

Specific comments:

1. Page 14, lines 11-19. The authors interpret the ion velocity shift in the x and y directions (shown in Figures 6j and 6o) as 'gyration in the increasing northward Bz'. However, this interpretation is inconsistent with the magnetic field configuration in Figure 6a, in which the magnetic field is in the earthward direction (with nearly zero Bz) at the location of the light blue virtual probe.

I would propose an alternative interpretation. Any duskward-moving ion at the virtual probe has its instantaneous gyro-center to the south of the probe, which indicates the possibility of Speiser-type meandering orbits. On the other hand, the dawnward-moving ions with gyrocenters further northward can only stay in the northern hemisphere. Therefore, the higher fluxes in the duskward rather than dawnward direction could be naturally understood by the higher density at locations closer to the neutral sheet.

In my understanding, the simulated ion distributions could be better explained by the model of Zhou et al. (2016, JGR, 'Understanding the ion distributions near the boundaries of reconnection outflow region'). As reconnection happens, the ions originally in the plasma sheet are picked up by the reconnection-associated Ey and Bz fields to move downstream away from the reconnection site. On top of this convective bulk motion, the pickup ions also keep meandering across the neutral sheet. These meandering ions must exhibit duskward and downstream directed velocities when they reach the off-equatorial boundary of the reconnection exhaust.

We agree with the Reviewer that the velocity distribution functions, observed in our simulations, may be interpreted in different ways. Particle tracing is necessary to find the

unique correct answer. The normal magnetic field component is, as correctly noted by the Reviewer, small but non-zero. Thus, ions of nearly thermal energy sense the positive Bz earthward of the X-line and negative Bz tailward of the X-line and gyrate accordingly. This, to our mind, explains the mirror-symmetric hook-like distributions in Fig. 6 h and j. The duskward shift of the phase space density seen in Fig. 6 m and o is caused by the reconnection electric field in the Y direction. We think that our explanation is not significantly different from that described by Zhou et al. [2016]: the ions are moving in the reconnection-associated Ey and Bz away from the X-line and duskward. We found the paper by Zhou et al. [2016] very instructive and added the citation on page 14, l. 16.

2. Page 16, First paragraph. The authors state that 'Fermi acceleration continues during the tailward convection of the plasma'. I don't get this picture, since my understanding of the Fermi acceleration is that requires two magnetic mirrors moving towards each other. But do we have magnetic mirrors tailward of the reconnection site?

We thank the Reviewer for valuable comment. In fact, we used the term "Fermi acceleration" mainly as a synonym for "parallel energization". Our simulations indicate that the energy of field-aligned beams in ion velocity distribution functions increases during tailward progression of the reconnected flux tube. That implies a mechanism of parallel energization. Again, because we cannot trace an individual particle in the Vlasiator code, we can only speculate that this energization is caused by the flux tube shortening, i.e., the Fermi acceleration process (e.g., Baumjohann and Treumann, Basic Space Plasma Physics, Revised Edition, 2012, p.32). It is seen from the magnetic field configuration shown in Figure 2 that there are X-lines tailward of the major X-line at -13 RE. These X-lines play a role of "magnetic mirrors". We added the necessary explanations on P. 16, L. 7-10.

Technical corrections:

1. Page 10, line 15: '12.5 R_E' should be '-12.5 R_E'.

The correction was made, thank you!

2. Figure 5, left panel: I don't quite understand the label of the horizontal axis, 'V_v perp B'.

It means velocity in V_perp direction, i.e., the direction of the ion bulk velocity component perpendicular to the instantaneous magnetic field in the grid cell where the vertual detector is placed. The left panel in Figure 5 shows the ion velocity distribution function cut in the plane perpendicular to the instantaneous magnetic field. The x-axis is along V-perp and the y-axis is along **VxB**. We have edited the Figure and added the necessary explanation to the text figure captions.

**Response to Reviewer #2**

The paper provides a comprehensive documentation of ion (proton) velocity distributions obtained within a 2D hybrid Vlasov simulation of global solar wind/magnetosphere interaction causing magnetotail reconnection and plasmoid ejection. This is the first self-consistent result of this kind obtained on a global realistic scale. The resulting distributions show good consistency with previously published Themis observations (supplemented by an additional new figure) and MHD/test particle simulations. This may be seen as both validation of the Vlasiator results and the non-self-consistent test particle results. I have only minor points of clarification.

Speiser orbits are particular orbits in magnetotail like fields with finite Bz that consist of quasi-adiabatic gyro motions outside the neutral sheet, turning into a meandering motion when the particle enters the central current sheet (together with an approximate half-gyration around the finite Bz), and become gyromotions again when the particle exits toward higher latitude on the same or the opposite side. It is not clear whether the author refer to such motion or just to the meandering part, which is an indication of non-adiabaticity. While distributions, such as in Fig. 6, are an indication of non-gyrotropy, it is not clear how they would indicate specifically Speiser type orbits.

We thank the Reviewer for a great question. In our study, we placed virtual detectors close to the equatorial plane (Bx=0). Therefore, we observe the meandering part of the Speiser trajectory. Particles there experience an acceleration in the dawn-to-dusk electric field that leads to an asymmetry in the phase space density (vy>0), that are visible in Figure 6 m, n, and o. Similar distributions were observed in simulations and observations and were attributed to the Speiser-type meandering motion (e.g., Nagai et al., 2015, doi:10.1002/2014JA020737; Hietala et al., 2015, doi:10.1002/2015GL065168). We interpreted the half-ring distributions in Fig. 6 m, n, and o as signatures of the meandering motion. We have added necessary explanations to the text (P. 14, l 21-25).

Page 2, line 11: The original term used by Liu et al is "dipolarizing" flux bundle, as is used also later in the paper. One might, however, argue that "dipolarizing" implies "increasing Bz," which probably was not intended by Liu and I would not object to leaving this as is.

Point taken. An increase in Bz is one of the criteria which were used by Liu et al (2013) to define "dipolarazing flux bundles" (DFB). Yet, one should distinguish DFBs, which are transient magnetic structures, and "dipolarizations", i.e., gradual, temporal increase in Bz. We have added the necessary clarification in the revised manuscript (P. 2, l. 11-14).

Page 3, line24: The previous sentence refers to Fermi acceleration, causing the field-aligned beams. Simply change: "This" to "The" and a bit later eliminate "thus." Also, the two effects are the same: adiabatic convection toward increasing B is, in the moving frame, the betatron effect.

We thank the Reviewer for the valuable suggestion, we have implemented it.

Page 5, line 25: Shouldn't this be a "cylinder" rather than a "sphere"?

Indeed, strictly speaking, the 2D geometry and the usage of a line dipole make it more exact to refer to the inner boundary as a perfectly conducting cylinder than a sphere. We have made the correction in the revised manuscript. Thank you!

*[Additional comment sent to the Editor on 12 April 2021]* First order Fermi acceleration, as discussed in Northrop's book, may consist of either type A, reflections between magnetic mirrors moving toward each other (as mentioned by Referee 1) or type B, crossing from one side of a moving curved magnetic field line to the other (slingshot effect). In collapsing field lines earthward of a reconnection site, the acceleration in shortening closed magnetic flux tubes may be interpreted in either way, particularly for electrons, which may bounce many times. On the tailward side type B may also apply. But I agree with the fact that parallel acceleration does not necessarily imply Fermi acceleration. The non-adiabatic Speiser type orbits lead to similar results.

We thank the Reviewer for the clarification. We have revised the text adding the explanations on Fermi types A and B acceleration mechanisms.